# Optimizing Pre-Training of Tabular Foundation Models by Shaping Geometry

**Humzah Merchant** [1] [2]    **Sriniketh Vangaru** [1]    **Randall Balestriero** [1]

## Abstract

Tabular foundation models (TFMs) are trained to solve new supervised learning problems in-context, despite not knowing the downstream table, feature distribution, or label rule during pre-training. This suggests that classifier TFMs might learn broad, well-spread intermediate representations that later layers can adapt to task-specific decision boundaries. Empirically, we do not observe this. We therefore use row- and column-wise Sketched Isotropic Gaussian Regularization (SIGReg) for transformer hidden states to directly shape representation geometry. Early-layer SI-GReg reorganizes the model across depth, producing better-distributed early embeddings while preserving later layers for task-specific decision structure. This improves convergence speed, classification performance, and seed-to-seed stability with little additional overhead.[1]

## 1. Introduction

Tabular foundation models (TFMs), such as TabPFN and TabICL (Hollmann et al., 2023; Grinsztajn et al., 2026; Qu et al., 2025), solve new prediction problems in-context: after being trained on synthetic data, they generalize to completely unseen tables, learning the relationship between features at inference time. This leads to a question about representation geometry: what kind of embedding space should a model learn to perform in-context learning?

We hypothesize that supervised TFM pretraining on many synthetic and randomly generated tables naturally produces broad, well-spread intermediate embeddings (Shwartz-Ziv et al., 2024) that later layers can adapt to task-specific decision boundaries. Isotropy is not theoretically desirable at later layers because isotropic embeddings may conflict with the clustered, linearly separable structure needed for

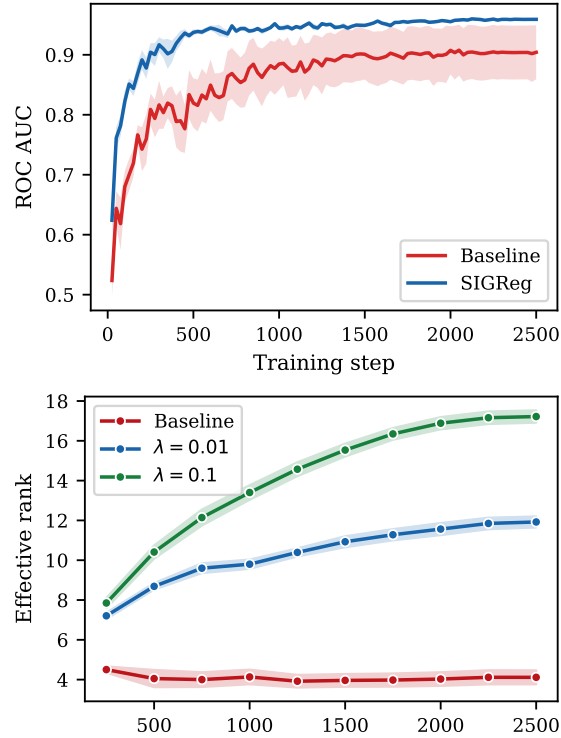

*Figure 1.* SIGReg improves training performance (top) by encouraging early-layer embeddings to be well dispersed (bottom.)

classification (Mickus et al., 2024). We test this hypothesis and find that it does not hold: although the model learns in-context prediction, its embeddings do not naturally approach a well-spread distribution over the course of training.

We therefore use Sketched Isotropic Gaussian Regularization (SIGReg) (Balestriero & LeCun, 2025) as a direct intervention on TFM hidden-state geometry. SIGReg improves classification performance, convergence speed, and seed-to-seed stability. Our main finding is that early-layer geometric regularization changes how the model uses the depth it has available. Linear probing (Alain & Bengio, 2018) and principal component analysis (Hotelling, 1933) visualizations suggest that over the course of training, the regularization creates a better-distributed early representation while leaving later layers free to form task-specific decision structure.

We make three contributions: (i) we show that supervised

[1]Brown University [2]University of Chicago. Correspondence to: Humzah Merchant <humzah_merchant@brown.edu>.

*Proceedings of the 2nd ICML Workshop on Foundation Models for Structured Data*, Seoul, South Korea. 2026. Copyright 2026 by the author(s).

[1]https://github.com/galilai-group/tabular-sigreg

TFM pretraining does not naturally produce isotropic Gaussian intermediate embeddings on `TabPFN` style training; (ii) we apply row- and column-wise SIGReg for tabular transformer hidden states; and (iii) we show that shaping early-layer geometry improves classification pretraining while reorganizing the learned representations across depth.

## 2. Method

### 2.1. Overview

We use `nanoTabPFN` (Pfefferle et al., 2025) which mimics the `TabPFN v2` (Hollmann et al., 2025) architecture. The model uses a feature and target encoder; each transformer stack applies attention by features (row) and then attention between datapoints (column). We train on provided synthetic tabular datasets.

We observe non-trivial variation across weight initialization and data ordering seeds; by performing 10 independent runs per experiment, we isolate the precise geometric effects of SIGReg. This approach follows established practices in other fields of machine learning, such as LLM research focusing on 1–8B parameter models before being applied at the trillion-parameter scale (Radford et al., 2019). Given the thousands of GPU hours for a single training run of TabPFN-v2, using it would have made it impractical. We recognize this as a limitation of the study. Our evaluation reports averages across multiple tabular datasets.

### 2.2. Gaussian regularizers

VICReg (Bardes et al., 2022) applies separate terms that encourage variance, covariance, and invariance among embeddings. We use VCReg (Zhu et al., 2024), which drops the invariance term. By discouraging covariance among embedding dimensions, VICReg encourages isotropy.

SIGReg (Balestriero & LeCun, 2025) is the evolution of VICReg, instead using a sliced Epps–Pulley statistic that integrates a closed-form characteristic-function distance to the standard normal distribution over many random 1D projections. SIGReg offers several advantages over its predecessor, including linear time complexity. We therefore direct our focus on SIGReg, though we demonstrate that the performance gains are nearly identical across methods.

### 2.3. Applying regularization

Let $\mathbf{H}^{(\ell)} \in \mathbb{R}^{B \times R \times C \times E}$ denote the hidden state at layer $\ell$ ($B$ tables, $R$ rows, $C$ columns, $E$ embedding dimension). Since there is no single canonical population of embeddings to regularize, we define two complementary views and regularize each independently. The *column view* groups the $R$ datapoint embeddings within each $(b, c)$ pair; the *row view*

groups the $C$ feature embeddings within each $(b, r)$ pair:

$$\mathbf{H}_{\text{col}}^{(\ell)} = \text{reshape}\Big( \text{perm}(\mathbf{H}^{(\ell)}, (0, 2, 1, 3)) \Big) \in \mathbb{R}^{BC \times R \times E}$$

$$\mathbf{H}_{\text{row}}^{(\ell)} = \text{reshape}(\mathbf{H}^{(\ell)}) \in \mathbb{R}^{BR \times C \times E}$$

Each view is passed through its own projection head $g_v$ : $\mathbb{R}^E \to \mathbb{R}^D$ ($v \in \{\text{c}, \text{r}\}$, two-layer MLP with BatchNorm (Ioffe & Szegedy, 2015)) so that the column and row regularizers can specialize independently. As empirically observed in **Appendix B.2.2**, these projectors are critical. SIGReg then evaluates per group and averages over the $BC$ (resp. $BR$) groups.

### 2.4. Training objective

The full training loss is

$$\mathcal{L} = (1 - \lambda_{\text{c}} - \lambda_{\text{r}}) \, \mathcal{L}_{\text{sup}} + \sum_{v \in \{\text{c}, \text{r}\}} \lambda_v \, R\Big( g_v(\mathbf{H}_v^{(\ell)}) \Big),$$

where $\mathcal{L}_{\text{sup}}$ is cross-entropy (classification) or bar-distribution NLL (regression). We use the $1 - \lambda$ formation to be consistent with LeJEPA (Balestriero & LeCun, 2025).

## 3. Results

### 3.1. TFMs do not naturally become well dispersed

First, we test the hypothesis that the training objective of TFMs naturally encourages them towards the isotropic Gaussian distribution. We use three measures of normality: the Sketched Epps-Pulley (Epps & Pulley, 1983) score (SIGReg without any projector), the effective rank of the embeddings, and the mean per-dimension standard deviation. By all metrics, the baseline model's embeddings become less Gaussian over the course of training, with **Figure 1** showing that the effective rank of the baseline model's embeddings after the first layer stays nearly constant across training. The other metrics, across all layers, are presented in **Appendix B.2.1**.

### 3.2. Cost of SIGReg

The cost of SIGReg approximately scales as $O(D)$, where $D$ is the embedding dimension of the model. We evaluate the overhead per step (measured in wall clock time) of SIGReg on an Nvidia A40 GPU using the TabPFN-2.5 setup and model dimensions, sweeping each hyperparameter. We consider one and two hidden layer projectors of shape

$$E \to 4E \to E/4 \quad \text{and} \quad E \to 4E \to 4E \to E/4$$

where $E$ is the embedding dimension.

We empirically verify this using a model with the same size and architecture as `TabPFN v2.5`. As shown in **Table 1**, the SIGReg computation imposes a fixed cost that scales

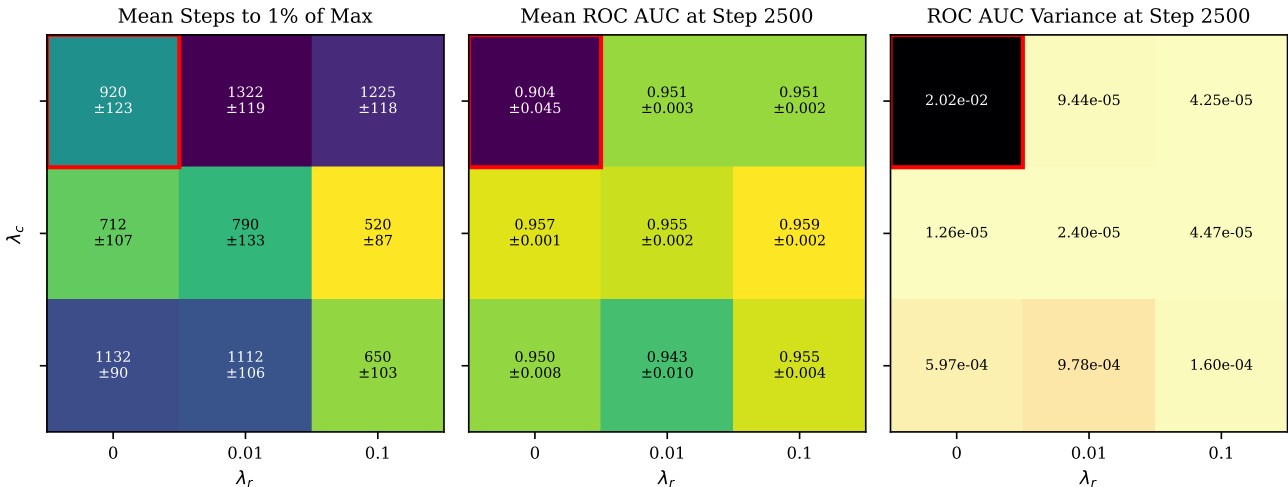

*Figure 2.* SIGReg improves all aspects of classification pretraining performance across values when doing a grid sweep over $\lambda_c$ and $\lambda_r$.

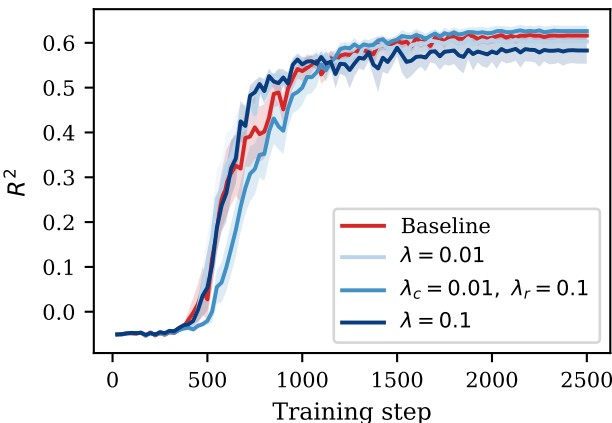

*Figure 3.* The theoretical case for why SIGReg is useful on classification **does not** apply to regression. As a result, regression serves as a control: since the benefits of SIGReg are limited, this demonstrates that it is useful beyond being a generic regularizer.

| Embedding Dimension | SIGReg cost (norm.) |
|---|---|
| 48 | 0.88 |
| 96 | 0.92 |
| 192 | 1 |
| 384 | 1.25 |
| 768 | 1.97 |

*Table 1.* Absolute added cost of SIGReg per step scales only with embedding dimension. See **Table 2** for complete results.

only with embedding dimension, not with model depth, FFN hidden dimension, or attention heads as seen in **Appendix B.1**. Concretely, SIGReg adds an approximately 10-15% overhead per step while reducing total steps to convergence by nearly half (**Figure 2**, left panel).

### 3.3. Applying SIGReg

**Classification.** We apply SIGReg on the first layer of the model for classification tasks and observe improvements in convergence speed, final performance, and seed-to-seed stability across a wide range of $(\lambda_c, \lambda_r)$ pairs (**Figures 1** and **2**). Results for training with the VCReg loss (Mialon et al., 2024) are given in **Appendix B.2.3**. In contrast, applying SIGReg to the second layer is increasingly harmful for training, consistent with the idea that enforcing isotropic geometry too close to the classifier can interfere with linearly separable task structure (Mickus et al., 2024).

**Regression.** Regression provides a useful contrast to classification because the desired representation geometry is different. If SIGReg's classification gains were primarily due to generic optimization benefits, such as stabilizing high-variance batches or improving conditioning, we would expect similar improvements on regression tasks as well. Instead, we observe only limited gains for regression (**Figure 3** and **Appendix B.3**). We visualize the natural and intervened embedding dispersions in Figure 14.

This difference is consistent with the structure of the two task families. Classification benefits from representations that support separable decision regions, whereas regression requires estimating a continuous numerical quantity. Useful regression representations should preserve metric or ordinal relationships between examples: points with nearby target values should remain nearby, or at least similarly ranked, in feature space, reflecting the assumed continuity of the input-output mapping (Gong et al., 2022). Thus, the weak regression results support our interpretation that SIGReg helps classification by shaping a useful early-layer geometry, rather than merely improving optimization stability.

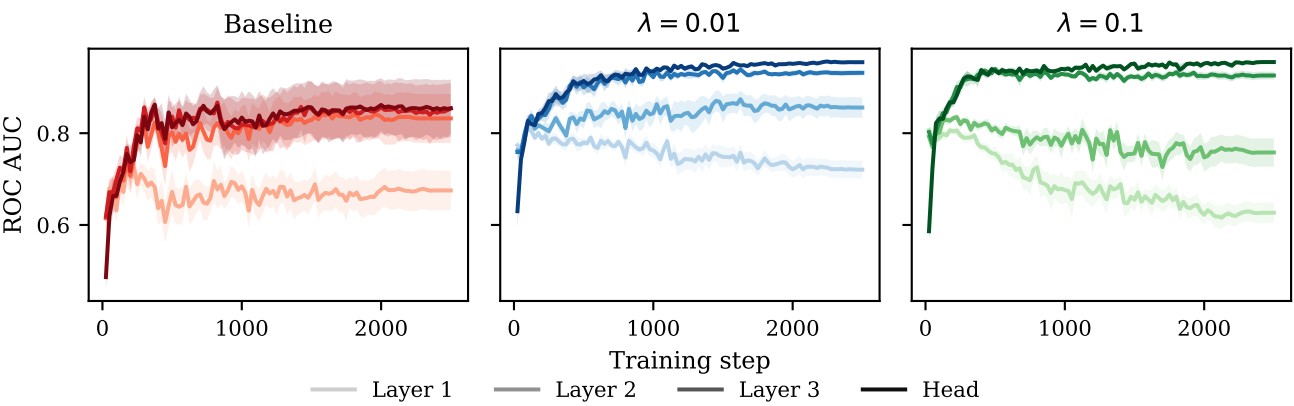

*Figure 4.* Applying a linear probe to the embeddings after each layer of the model. SIGReg changes how the model uses depth, with the embeddings becoming more linearly separable after each layer.

### 3.4. Embedding Geometry

**Linear probing.** We apply linear probes (Alain & Bengio, 2018) to the embeddings after each of the layers across training. **Figure 4** shows three noteworthy patterns:

- The baseline model's second layer's performance is nearly as good as the final classifier head

- With SIGReg, the linear separability of embeddings outputted by earlier layers gets worse over the course of training

- With SIGReg, the linear separability of the embeddings after the third layer stops improving early into training but the head continues to improve in performance.

**PCA visualizations.** We apply PCA (Hotelling, 1933) to the embeddings after each layer and plot the first two principal components. Color indicates class, while marker type indicates train/test split (dot = train, X = test). **Figure 5** shows the circles dataset averaged over all 10 seeds at step 2500 and additional datasets are shown in **Appendix B.2.4**. Across all three, we see a similar pattern: the first two layers offer little separation between the two classes in the baseline model, while the models with SIGReg scatter embeddings.

Together, these results suggest that SIGReg redistributes the work across layers: early layers produce broadly distributed representations (even though they are not necessarily linearly separable), leaving later layers free to form task-specific decision structure, which is a more efficient use of model depth than the baseline achieves.

## 4. Conclusion

We demonstrate that tabular foundation models, despite being trained to generalize across tasks, do not naturally produce isotropic Gaussian embeddings. However, applying SIGReg at an early transformer layer during pretraining

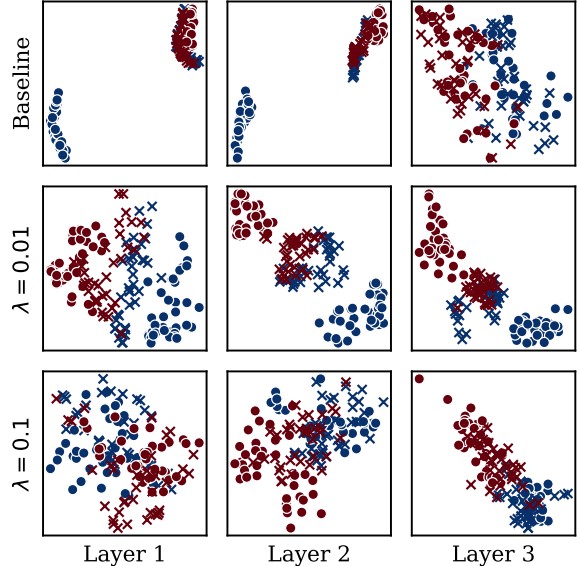

*Figure 5.* Embeddings projected into first two principal components shows that SIGReg changes how the first two layers are used, using the first two layers to scatter the embeddings, allowing for stronger seperability later.

improves downstream classification performance with little added overhead. These findings suggest that explicitly shaping intermediate representation geometry during pretraining is a promising and theoretically grounded direction for improving tabular foundation models. These results also motivate further exploration of other embedding-space methods, e.g. JEPA (Thimonier et al., 2025; Chen et al., 2023).

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

## A. Related Work

**Tabular foundation models and representations.** TabPFN introduced prior-data fitted transformers that solve new tabular prediction tasks in-context after pretraining on synthetic tasks (Hollmann et al., 2023; 2025). Recent work has begun to analyze the internal representations and mechanisms of these models: (Gupta et al., 2026) probe TabPFN hidden states and find that linear coefficients, intermediate arithmetic quantities, and final-answer signals are decodable across layers, suggesting structured internal computation. (Ye et al., 2025) analyze how TabPFN v2 handles heterogeneous attribute spaces through randomized attribute tokens and in-context inference of attribute relationships, while also documenting limitations on high-dimensional, many-category, and large-scale tasks. Model reports for TabPFN-2.5 further emphasize architectural and scaling changes for larger tables (Grinsztajn et al., 2026). We extend this line of literature by directly regularizing the geometry of TFM hidden states during pretraining.

We encourage readers who are unfamiliar with TFMs to read the excellent explanation and code of nanoTabPFN (Pfefferle et al., 2025).

**Representation regularization.** Our work builds on regularizers that prevent collapse and encourage useful embedding geometry. The Sketched Isotropic Gaussian Regularization (SIGReg) loss term targets an isotropic Gaussian embedding distribution using a sketched characteristic-function objective, motivated by task-agnostic downstream prediction (Balestriero & LeCun, 2025). VICReg instead uses variance and covariance penalties to avoid collapse in self-supervised learning (Bardes et al., 2022); its supervised variant, often called VCReg, removes the invariance term and regularizes representations to maintain high variance and low covariance (Mialon et al., 2024; Zhu et al., 2024). We adapt these ideas to tabular transformers by applying row- and column-wise regularization to intermediate hidden states.

## B. Additional Results

### B.1. Additional wall-clock costs when applying SIGReg

The wall-clock time of computing the SIGReg loss when sweeping over all model hyperparameters is shown in **Table 2**. SIGReg has approximately the same absolute cost across all hyperparams except embedding dimension.

### B.2. Classification

#### B.2.1. NORMALITY OF EMBEDDINGS

**Figure 6** shows the results of measuring normality across all metrics and all layers. The layer index corresponds to the transformer layer which output the embeddings being measured; the SIGReg loss was calculated on the first layer during training (when not using the baseline model) for all plots.

#### B.2.2. TURNING OFF THE PROJECTOR

We test removing the projector and applying the SIGReg loss directly to the embeddings. We find that this greatly reduces performance (**Figures 7** and **8**).

#### B.2.3. USING THE VCREG LOSS

We test the VCReg loss terms and find that they perform equally as well as SIGReg, as shown in **Figures 9** and **10**. In order to compute the loss separately on row and column embeddings as we do with SIGReg, we first fix the hyperparameters from VICReg (Bardes et al., 2022) at $\mu = 25$ and $\nu = 1$, and then effectively remove the invariance term by setting its coefficient $\lambda = 0$. We additionally use a projector in the same manner as with SIGReg, though the projected embeddings for both views are reshaped to $(B \times R \times C, E)$ before computing the variance and covariance losses.

#### B.2.4. EMBEDDING GEOMETRY USING PRINCIPAL COMPONENT ANALYSIS

Decision boundaries for the embeddings outputted by each layer when sweeping across $\lambda$ are shown for the linear and moons datasets in **Figures 11** and **12**, respectively.

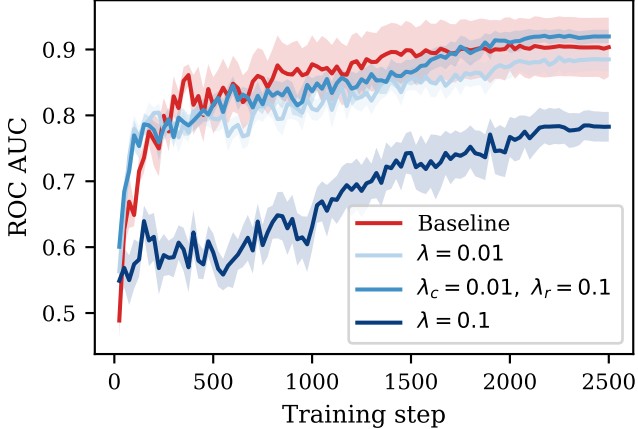

*Figure 6.* All normality metrics across all layers for classification with SIGReg applied after Layer 1.

*Figure 7.* Training with SIGReg for classification without using a linear projector.

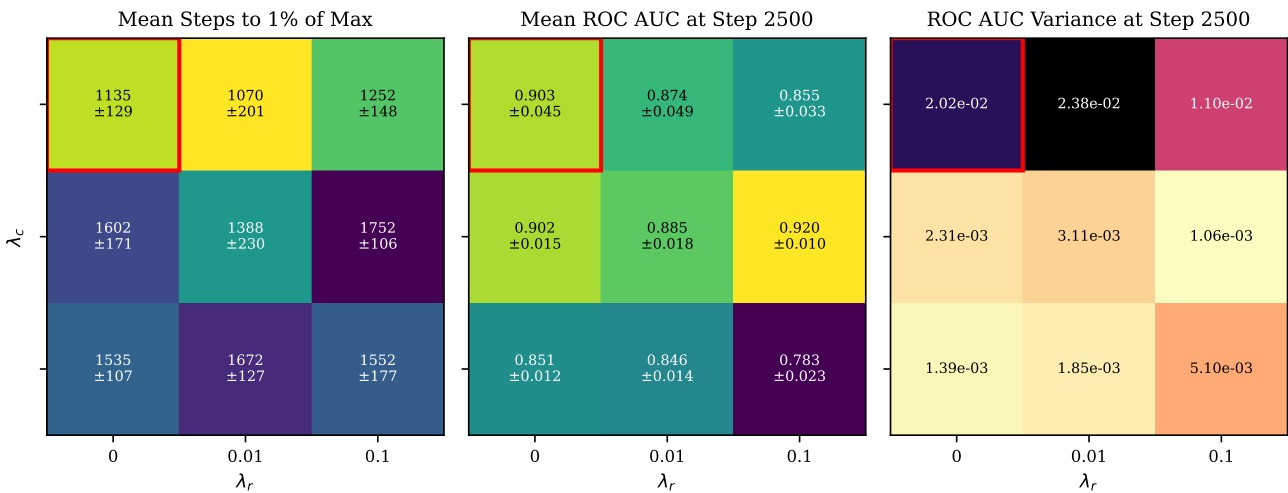

*Figure 8.* Heatmaps when training with SIGReg for classification without using a linear projector.

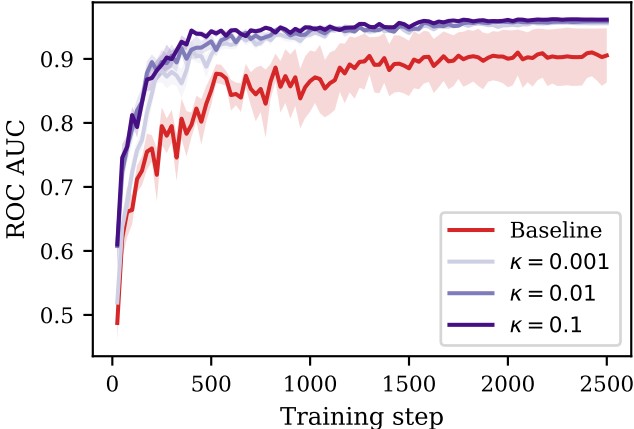

*Figure 9.* VCReg on classification.

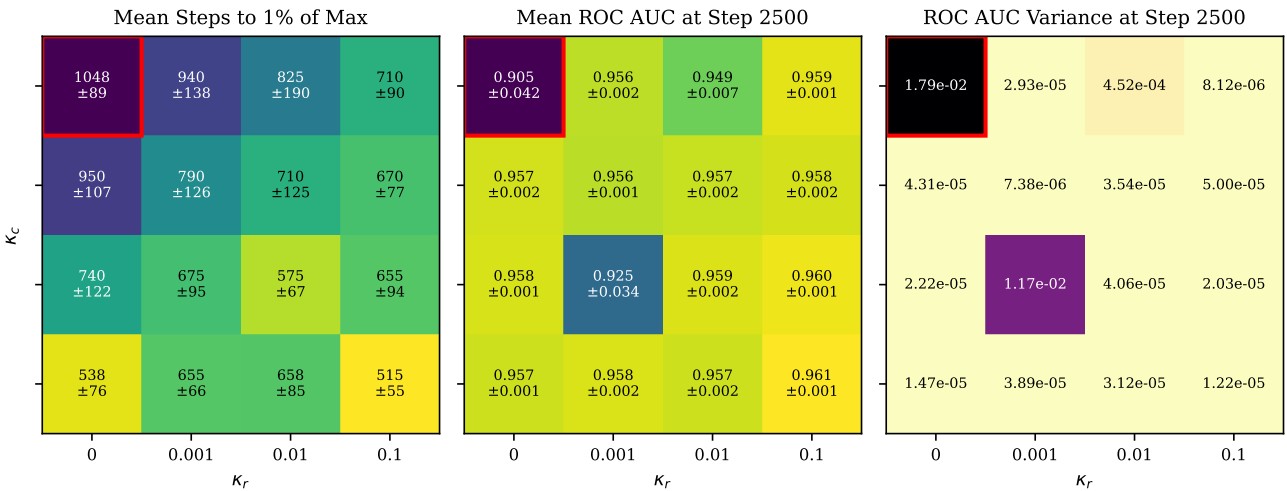

*Figure 10.* Heatmaps of evaluation metrics for VCReg on classification.

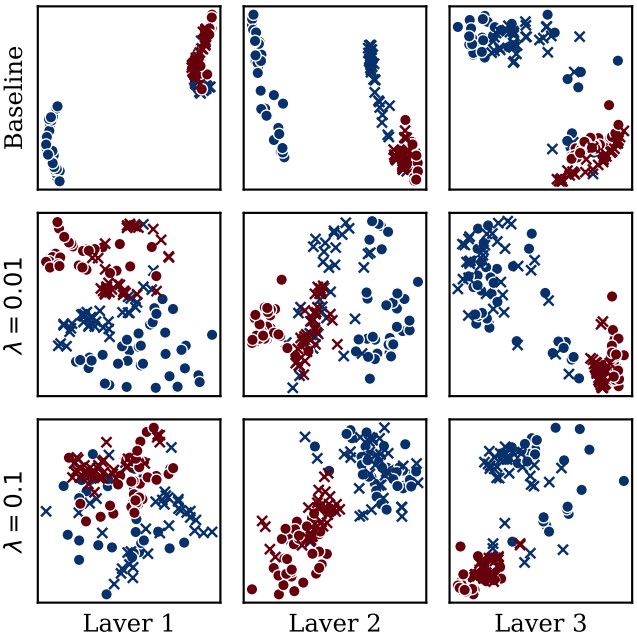

*Figure 11.* Decision boundaries for the linear dataset.

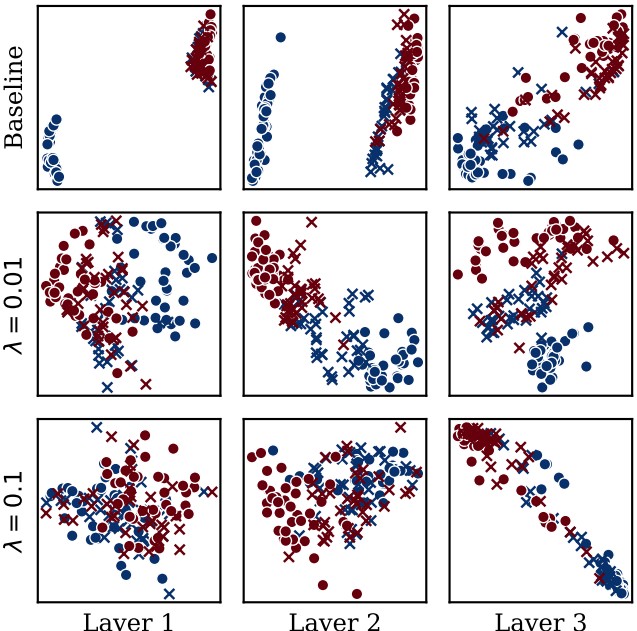

*Figure 12.* Decision boundaries for the moons dataset.

| Layers | 1-layer SIGReg cost (norm.) | 2-layer SIGReg cost (norm.) | Total params | 1-layer overhead | 2-layer overhead |
|---|---|---|---|---|---|
| 6 | 1 | 1 | 2.75M | 47.3% ± 0.0% | 58.3% ± 0.1% |
| 12 | 1 | 0.99 | 5.42M | 23.8% ± 0.1% | 29.1% ± 0.0% |
| 24 | 1 | 1 | 10.77M | 11.9% ± 0.0% | 14.7% ± 0.0% |
| 36 | 0.98 | 0.99 | 16.12M | 7.8% ± 0.1% | 9.8% ± 0.0% |
| 48 | 1 | 0.99 | 21.47M | 6.0% ± 0.0% | 7.3% ± 0.0% |

*(a)* Sweep over model depth (other base hyperparameters fixed; default row underlined).

| Embedding dim D | 1-layer SIGReg cost (norm.) | 2-layer SIGReg cost (norm.) | Total params | 1-layer overhead | 2-layer overhead |
|---|---|---|---|---|---|
| 48 | 0.88 | 0.75 | 1.37M | 20.6% ± 0.1% | 21.6% ± 0.1% |
| 96 | 0.92 | 0.82 | 3.62M | 16.7% ± 0.0% | 18.3% ± 0.0% |
| 192 | 1 | 1 | 10.77M | 11.9% ± 0.0% | 14.7% ± 0.0% |
| 384 | 1.25 | 1.63 | 35.69M | 8.1% ± 0.0% | 13.0% ± 0.0% |
| 768 | 1.97 | 3.65 | 127.99M | 5.5% ± 0.1% | 12.7% ± 0.0% |

*(b)* Sweep over embedding dimension D (other base hyperparameters fixed; default row underlined).

| FFN hidden dim | 1-layer SIGReg cost (norm.) | 2-layer SIGReg cost (norm.) | Total params | 1-layer overhead | 2-layer overhead |
|---|---|---|---|---|---|
| 96 | 1 | 1 | 8.05M | 12.7% ± 0.0% | 15.6% ± 0.0% |
| 192 | 1 | 1 | 8.96M | 12.3% ± 0.0% | 15.1% ± 0.0% |
| 384 | 1 | 1 | 10.77M | 11.9% ± 0.0% | 14.6% ± 0.0% |
| 768 | 1.01 | 1 | 14.39M | 11.2% ± 0.0% | 13.7% ± 0.0% |
| 1536 | 1.01 | 1 | 21.64M | 9.7% ± 0.0% | 11.8% ± 0.0% |

*(c)* Sweep over FFN hidden dimension (other base hyperparameters fixed; default row underlined).

| Heads | 1-layer SIGReg cost (norm.) | 2-layer SIGReg cost (norm.) | Total params | 1-layer overhead | 2-layer overhead |
|---|---|---|---|---|---|
| 2 | 0.98 | 1 | 10.77M | 14.1% ± 0.0% | 17.6% ± 0.0% |
| 3 | 1 | 1.01 | 10.77M | 13.9% ± 0.0% | 17.0% ± 0.0% |
| 6 | 1 | 1 | 10.77M | 12.0% ± 0.1% | 14.6% ± 0.0% |
| 12 | 1 | 1.01 | 10.77M | 9.5% ± 0.0% | 11.8% ± 0.1% |
| 24 | 1 | 1.01 | 10.77M | 6.7% ± 0.0% | 8.3% ± 0.0% |

*(d)* Sweep over number of heads (other base hyperparameters fixed; default row underlined).

*Table 2.* SIGReg wall-clock overhead vs. each architecture hyperparameter, holding the others at the TabPFN-2.5/2.6 base config ($D$=192, FFN=384, 24 layers, 6 heads). Errors are given by ±1 SE over 10 timed steps.

## B.3. Regression

Further evaluation metrics for SIGReg when training and testing on regression tasks are shown in **Figures 13** and **14**.

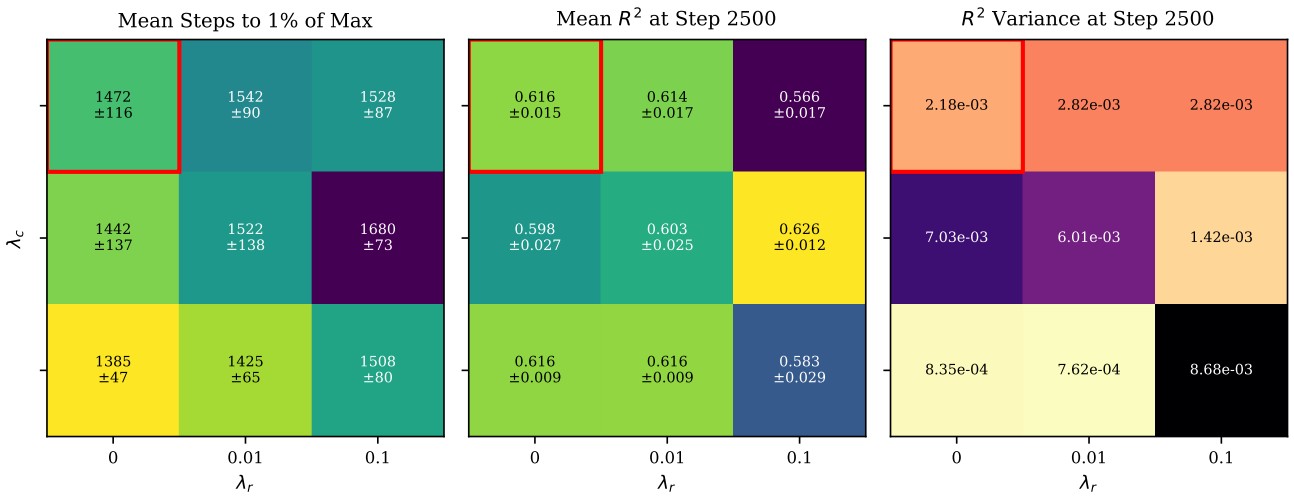

*Figure 13.* Heatmaps of evaluation metrics for SIGReg on regression.

*Figure 14.* All normality metrics across all layers for regression with SIGReg applied after Layer 1.

