# OpenReview forum: "Optimizing Pre-Training of Tabular Foundation Models by Shaping Geometry"
_ICML.cc/2026/Workshop/FMSD — FMSD @ ICML 2026 Poster_

### Official Review · Reviewer_dSSP · 2026-05-15
**Unclear of the actual insights, but interesting future avenues**

**Rating:** 6
**Confidence:** 4

**Review:**

**Summary**: the authors adapt a regularization technique to shape embedding geometry of tabular representations

Recommending to accept because I believe the answer to the question "Would at least some individuals in this workshop’s audience be interested in knowing the findings of this paper?" is roughly yes and workshops are opportunities to expand preliminary works and get feedback. However, this paper needs significant work.

Namely, I disagree with the core motivation and insight of the first contribution of the paper.  "Because the isotropic Gaussian is theoretically optimal for downstream classification tasks..." is doing a lot of heavy lifting and is at best imprecise, if not misleading. LeJepa paper explicitly states "Together, these results establish the isotropic Gaussian distribution as the optimal design to minimize the **worst-case risk** of a foundation model across downstream tasks." The LeJepa paper is using a very conservative notion of risk here, and it is pretty obvious that a model trained by empirical risk minimization will not conform to a minimax result, in general. In that sense, the first contribution is rather unsurprising and the claims are fairly broad, especially for only trying out one architecture. Even if we agreed on the scientific angle of the question, you do not really show anything for "supervised TFM pretraining" in general, you use one smaller model.

The higher-level question is interesting: of the foundation models that exist, does their performance ordering rely on their ability to organize early layers isotropically? But this would require testing multiple architectures and likely different regularization strategies. If this is going to be presented at the workshop, the authors should strike (i) as a contribution.

Next, for contribution (ii). The paper frequently uses "we introduce..." but this language, to me, seems pretty misleading as this specific regularization has already been introduced in previous work. "We utilize SIGReg in the context of tabular data..." is more appropriate here (as you used once in the intro). I also question the specific utility of SIGReg, in addition to a baseline w/o regularization, the authors should probably have included other generic regularization strategies here. The question of which layers should be regularized is interesting, but that it is due to SIGReg-specific structure here is not entirely addressed, as Fig 7 shows. You have two axes of variation but not cleanly disentangle them.

**Justification of score** this has clear relevance for the workshop and I think the discussions around its high level questions warrant its acceptance. However, the specific contributions of this work are still quite preliminary and need continued work

Also, random note: your anonymous code seems to have a README with the git ssh link still in there? git clone git@github.com:galilai-group/latable.git. If you're going to include this, please be careful for future work. A random username is fine but that it clearly contains "galilai-group" might get you burned in future work

---

### Official Review · Reviewer_VL7F · 2026-05-20
**Shaping embedding geometry of tabular foundation models by enforcing isotropic Gaussian.**

**Rating:** 7
**Confidence:** 3

**Review:**

### Summary
The authors evaluate a strategy to shape embedding geometry in tabular foundation models, by specifically enforcing an isotropic Gaussian in early layers embeddings. This shows improvements in downstream predictive performance on classification tasks as well as faster convergence rates.

### Strengths
All three contributions are clearly motivated and build naturally on one another.

The paper is well-written, with each experimental step following a logical progression.

The contrast on regression tasks is good argument for highlighting the classification gains as geometric rather than optimization. Also the much improved seed-to-seed stability is an interesting additional contribution.

### Areas for Improvement
While the authors discuss that applying SIGReg to later layers is increasingly harmful, cases where the approach hurts in early layers are not properly discussed. In particular, it remains unclear how sensitive the gains are to dataset characteristics such as feature correlation structure or dataset size. For instance, when features are strongly correlated, the enforced isotropy in the early layers might be harmful rather then beneficial. A small ablation or discussion along these lines would help to better understand the limitations.

Do the results for regression suggest that enforcing isotropic Gaussian only really has a benefit for classification tasks and that for regression task TFMs already learn the optimal representation geometry? Or that we need fundamentally different pre-training strategies for regression models in general? A discussion on what good regression geometry should look like would be a very interesting extension to this work.

### Detailed Comments
See detailed discussions in the Areas for Improvement.

As an additional comment: Not really a weakness of the paper as it is also specifically acknowledged as limitation, but I am still wondering whether the gains achieved using SIGReg translate to larger scale models. An ablation on a larger model than NanoTabPFN would be a very insightful experiment towards understanding whether the approach generalizes to large scale training of current SOTA TFMs.

### Score
The paper provides good insights into shaping representations of tabular foundation models and its benefits on classification tasks. The paper is fitting the audience of the workshop and well structured. I am happy to argue for a score of 7 and hope that the comments help the authors to improve their work.

---

### Official Review · Reviewer_8WcM · 2026-05-22

**Rating:** 6
**Confidence:** 3

**Review:**

**Summary**

The paper studies representations in TFMs and demonstrates that encouraging early layer representations to be closer to the isotropic gaussian distribution (via the SIGReg loss) is beneficial for classification tasks (evaluating in a NanoTabPFN regime)

**Strength**

The paper proposes a novel and potentially interesting extension for TFM training procedures. The analysis and the difference between regression and classification task performance is intriguing. The performance boosts in classification seem significant (albeit in a toy-ish setup)

**Areas for improvement**

For sure, it would be interesting to see whether this method scales to more performant TFMs closer to the SoTA scale.

The reasoning that regression tasks do not benefit from better representation geometry is interesting, but could be expanded upon. Is it really the case (on how many datasest?), why so, could we probe the hypothesis more directly? What if we've used a binning-based approach to regression or quantile loss (the one used in TabICLv2), would it result in SIGReg like losses helping?

The paper currently misses some details regarding the setup, I infer that the datasets used for evaluation are the same as in NanoTabPFN? This should be specified in some way in the paper for completeness.

**Justification**

I think this work fits the workshop well, the score below 7 is just from the minor omissions in the paper, regarding the setup (datasets used for evaluation)